

# Abnormal mechanical stress on bicuspid aortic valve induces valvular calcification and inhibits Notch1/NICD/Runx2 signal

Guangzhou Li[1,*], Na Shen[2,*], Huifang Deng[1], Yixuan Wang[1], Gangcheng Kong[3], Jiawei Shi[1], Nianguo Dong[1] and Cheng Deng[1]

[1] Department of Cardiovascular Surgery, Union Hospital, Tongji Medical College, Huazhong University of Science and Technology, Wuhan, China

[2] Department of Breast and Thyroid Surgery, Union Hospital, Tongji Medical College, Huazhong University of Science and Technology, Wuhan, China

[3] Department of Hepatobiliary and Pancreatic Surgery, the First Affiliated Hospital, School of Medicine, Zhejiang University, Hangzhou, China

[*] These authors contributed equally to this work.

Corresponding author
Cheng Deng,
cheng_deng@hust.edu.cn

## ABSTRACT

**Background**. Bicuspid aortic valve (BAV) is a congenital cardiac deformity, increasing the risk of developing calcific aortic valve disease (CAVD). The disturbance of hemodynamics can induce valvular calcification, but the mechanism has not been fully identified.

**Methods**. We constructed a finite element model (FEM) of the aortic valve based on the computed tomography angiography (CTA) data from BAV patients and tricuspid aortic valve (TAV) individuals. We analyzed the hemodynamic properties based on our model and investigated the characteristics of mechanical stimuli on BAV. Further, we detected the expression of Notch, NICD and Runx2 in valve samples and identified the association between mechanical stress and the Notch1 signaling pathway.

**Results**. Finite element analysis showed that at diastole phase, the equivalent stress on the root of BAV was significantly higher than that on the TAV leaflet. Correspondingly, the expression of Notch1 and NICH decreased and the expression of Runx2 elevated significantly on large BAV leaflet belly, which is associated with equivalent stress on leaflet. Our findings indicated that the root of BAV suffered higher mechanical stress due to the abnormal hemodynamic environment, and the disturbance of the Notch1/NICD/Runx2 signaling pathway caused by mechanical stimuli contributed to valvular calcification.

## INTRODUCTION

Bicuspid aortic valve (BAV) is a congenital cardiac deformity, occurring in 1–2% of population worldwide and increasing the risk of developing valve leaflet calcification, aortic stenosis and aortic aneurysm (*Salcher et al., 2016*). Compared with individuals with tricuspid aortic valve (TAV), BAV patients are more likely to develop calcific aortic valve disease (CAVD) at early age (*Otto, 2002*; *Kong et al., 2017*). Studies have shown that

the leaflets from BAV individuals develop calcification much more readily due to the disturbance of hemodynamics (*Balachandran, Sucosky & Yoganathan, 2011*; *Lavon et al., 2021*). The deformity of aortic valve can cause abnormal flow dynamics and thereby leading to the dysfunction of endothelial cells, inflammation and valvular calcification (*Wang et al., 2021b*). Thus, investigating mechanical stimuli of BAV is crucial to shed light on the pathogenesis of CAVD.

The hemodynamic environment of aortic leaflets is intricate, containing pressure, shear stress and strain force during cardiac cycle (*Kazik et al., 2021*). Finite element analysis (FEA) can provide detailed information of morphological and mechanical parameters in the aortic root, which is an effective method to evaluate the hemodynamic changes in BAV patients (*Gomez et al., 2020*; *Nappi et al., 2021*). The fundamental theory is to discretize a continuous structure and divide it into finite elements connected to each other by nodes. Based on the parameters on the nodes of finite elements, a set of algebraic equations were formulated to describe the characteristics of these elements. In the 1970s, FEA was utilized to study the function of the aortic leaflet, and over the last few decades, FEA has proved to be a scientific, realistic and feasible method to analyze the mechanical stimuli of aortic valve. FEA is widely used in the research of cardiovascular diseases, such as heart valve diseases, coronary artery diseases, aortic aneurysm *etc.* (*Cataloglu, Clark & Gould, 1977*; *Pasta et al., 2021*). In our study, we constructed a finite element model of aortic valve based on the computed tomography angiography (CTA) data and compared the difference of hemodynamic parameters between BAV patients and TAV individuals.

There is growing evidence indicating the abnormal hemodynamic and biomechanical environment in BAV patients may contribute to the pathogenesis of CAVD. The disturbance of mechanical stress on aortic valves is thought to influence the function of valvular interstitial cells (VICs) and valvular endothelial cells (VECs), eventually leading to calcification. The presence of osteoblasts in calcific aortic valve suggests that the valve calcification is accompanied with lipoprotein deposition and chronic inflammatory response which is similar to atherosclerotic lesion (*Lerman, Prasad & Alotti, 2015*; *Peeters et al., 2018*; *Liu et al., 2022*). Current studies have found that the Notch1/NICD/Runx2 signaling pathway participates in the process of CAVD (*Marracino et al., 2021*; *Yan et al., 2022*). The Notch signaling pathway plays an important role in heart valve development and maintaining tissue homeostasis (*Wang et al., 2021b*). The activation of the Notch receptor induces the release of the Notch intracellular domain (NICD) and the translocation of NICD to nucleus is associated with the activation of target genes. The Notch1 mutations are risk factors for congenital BAV and acquired CAVD in humans (*Garg et al., 2005*). *White et al. (2015)* found that shear stress could activate the expression of Notch1 in VECs, downregulating the expression of the osteoblast-like gene. *Bosse, Mathieu & Pibarot (2008)* found that the transformation of VICs to Runx2 positive osteoblastic-like cell was associated with valve calcification. Thus, Notch1 signals might be disturbed by abnormal mechanical stress in BAV patients and play important role in the development of valvular calcification.

In our study, based on our finite element model, we found that the mechanical stimuli on the root of a large BAV leaflet was greatest among other parts of BAV and TAV. We

**Table 1 Clinical characteristics of four volunteers for the construction of finite element model.**

| Volunteer | 1 | 2 | 3 | 4 |
|---|---|---|---|---|
| AV type | BAV | BAV | TAV | TAV |
| Gender | Male | Male | Male | Male |
| Height (m) | 1.72 | 1.69 | 1.65 | 1.72 |
| Weight (kg) | 72 | 67 | 69 | 74 |
| BMI (kg/m$^2$) | 24.337 | 23.459 | 25.344 | 25.014 |
| LV (cm) | 4.5 | 4.6 | 4.5 | 4.3 |
| EF (%) | 60 | 70 | 70 | 70 |
| AAO (cm) | 3.2 | 3.1 | 2.8 | 3 |
| AV function | Normal | Normal | Normal | Normal |

**Notes.**

AV, aortic valve; BMI, body mass index; LV, left ventricle; EF, ejection fraction; AAO, ascending aorta inner diameter.

further collected the BAV and TAV samples and found that the Notch1 signaling pathway was disturbed at the belly of the large BAV leaflet, which was in accordance with the findings in FEA.

## MATERIAL AND METHODS

### Patient recruitment and sample collection

Two bicuspid aortic valve patients and two tricuspid aortic valve volunteers were enrolled in the construction of finite element model. BAV was diagnosed based on the cardiac ultrasonic examination (*Michelena et al., 2014*). The clinical characteristics of all participants were similar, including the height, weight, BMI, and the valves were functioning properly without regurgitation or stenosis based on cardiac ultrasonography (Table 1).

The BAV samples were collected from 22 BAV patients who underwent aortic valve replacement and the TAV samples were collected from 17 end-stage dilated cardiomyopathy patients who received donor heart from June 2016 to December 2016 at the Department of Cardiovascular Surgery of Wuhan Union Hospital. The BAV was diagnosed by direct observation during surgery to exclude the pseudobivalvular deformity, combined with echo report. The clinical demographic data were shown in Table 2.

### Consent details

In accordance with the Declaration of Helsinki, the study was approved by review boards of Union Hospital and Tongji Medical College (IORG0003571). Written informed consents were obtained from all participants at their inclusion in the study.

### Antibodies and primers

Antibodies for immunoblot in this study included the following: anti-Notch1 (ab280898; Abcam, Cambridge, UK), anti-Runx2 (ab114133; Abcam), anti-NICD (ab52301; Abcam), anti-GAPDH (ab8245; Abcam). Primers for qPCR in this study included the

**Table 2 Clinical demographic data of BAV and DCM patients.**

| Characteristics | Patients with BAV & calcification | Patients with DCM | *P* value |
|---|---|---|---|
| Patients (*n*) | 22 | 17 | |
| Age (y) | 50.55 | 44.53 | |
| Gender, Male (%) | 15 (68.2%) | 12 (70.6%) | 0.872 |
| Risk Factors | | | |
| Hypertension | 5 (22.7%) | 2 (11.8%) | 0.376 |
| Hyperlipidaemia | 4 (18.2%) | 3 (17.6%) | 0.966 |
| Diabetes mellites | 1 (4.5%) | 0 (0%) | 0.373 |
| Valvular dysfunction (aortic/mitral) | 22 (100%) | 6 (35.3%) | 0 |
| Hyperinflammatory responses (CRP/PCT) | 4 (18.2%) | 4 (23.5%) | 0.682 |
| ECG parameters | | | |
| LVEF (%) | $61.1 \pm 1.58$ | $28.2 \pm 2.4$ | <0.0001 |
| Transvalvular gradient (mmHg) | $62.5 \pm 30.4$ | 0 | <0.0001 |
| Diameter of annulus (cm) | $2.6 \pm 0.4$ | $2.0 \pm 0.6$ | 0.0006 |
| Diameter of ascending aorta (cm) | $4.6 \pm 0.5$ | $2.9 \pm 0.4$ | <0.0001 |
| Diameter of aortic sinus (cm) | $4.2 \pm 0.5$ | $2.7 \pm 0.6$ | <0.0001 |
| Transvalvular velocity (m/s) | $3.8 \pm 1.0$ | $1.4 \pm 0.4$ | <0.0001 |
| Medications | | | |
| Statins | 2 (9.1%) | 1 (5.9%) | 0.709 |
| ACEI/ARB | 3 (13.6%) | 3 (17.6%) | 0.731 |
| $\beta$-blockers | 5 (22.7%) | 4 (23.5%) | 0.953 |

**Notes.**

Data were shown as mean $\pm$ SD.

BAV, Bicuspid Aortic Valve; DCM, Dilated Cardiomyopathy; CRP, C-reactive protein; PCT, Procalcitonin; LVEF, Left Ventricular Ejection Fraction; ACEI, Angiotensin Converting Enzyme Inhibitor; ARB, Angiotensin Receptor Blocker.

following: Notch1 forward 5′-GCGACAACGCCTACCTCTG-3′, Notch1 reverse 5′-AAGCCATTGATGCCGTCC-3′, Runx2 forward 5′-GGACGAGGCAAGAGTTTCAC-3′, Runx2 reverse 5′-GAGGCGGTCAGAGAACAAAC- 3′, NICD forward 5′-CTATAGACGACATTGACGAGTGTGAC-3′, NICD reverse 5′-TGTAGAATTCAGAGGACAGTTC-3′, GAPDH forward 5′-AATCCCATCACCATCTTCCAG- 3′, GAPDH reverse 5′-GAGCCCCAGCCTTCTCCAT-3′.

## Establishment of the aortic valve geometric model

The aortic valve geometric model was established by a previously described method with modification (*Jernigan et al., 2007*; *Nappi et al., 2020*). Briefly, the ascending aorta CTA images in the DICOM format were imported into Mimics 10.01 software (Materialise, Leuven, Belgium). Then, based on the tissue density of cardiac valve, set the range of threshold between 329 to 560 Hounsfield units, and 3D reconstruction was performed using 'Calculate 3D' to obtain the geometric model of the aortic valve and ascending aortic arch. Next, the 3D solid model of aortic leaflets was built through meshes using the Remesh module in Magics.

## Parameters of the aortic leaflet model

Fibrous reinforcement material properties were utilized to model aortic valve in FEM module of the Mimics software based on the grayscale of CTA images. The ascending aorta was set as a large vessel with a uniform thickness of 2.3 mm and described by a linear hexahedral element. The aortic valve was set as thin tissue and described by a 4-node shell unit (*Kim et al., 2007*). The remaining tissues such as myocardium were set as isotropic linear elastic materials and the material properties were set as following: Young modulus was 2MPa, Poisson ratio was 0.3, tissue density was 1.1 g/cm$^3$. The response of leaflet to mechanical stimuli was considered to be highly elastic, isotropic and incompressible.

## Finite element analysis

The files of the aortic valve solid model were imported into ANSYS software (Materialise, Leuven, Belgium) for finite element analysis. The left ventricular end-diastolic pressure was set as 80mmHg, and the friction coefficient of aortic valve was set as 0.05. The finite element parameters of aortic valve were set according to isotropic, uniform continuous linear elastic material. The cardiac cycle was set as 0.8 s and we analyzed the equivalent stress on aortic valve and the deformation of leaflet during one cardiac cycle.

## Alizarin red stain

The aortic valve excised at the time of surgery was placed in a container filled with cold sodium chloride physiological solution, then fixed in 4% paraformaldehyde, dehydrated, and subsequently embedded in paraffin. Alizarin red stain was performed in paraffin sections of aortic valve tissues. After incubation with 0.2% alizarin red solution for 30 min, excess dye was removed by washing with distilled water. The images were captured using optical microscope (Olympus, Tokyo, Japan), and the calcification nodules appeared orange/red.

## Immunoblot analysis

Specimen from aortic valves were cut into pieces and smashed with liquid nitrogen. Protein extracted from aortic valve was then lysed in RIPA Lysis (Beyotime, Jiangsu, China) and Extraction Buffer (Beyotime) containing phenylmethanesulfonyl fluoride (PMSF, Beyotime) for 30 mins on ice. After 12,000 g centrifugation for 10 mins at 4 °C, the supernatants were then transferred to new Eppendorf tubes. The concentration of protein was measured using the BCA method and whole cell lysates were then subjected to western blot using standard procedures.

## RNA isolation, cDNA synthesis and RT-PCR

Specimens from aortic valves were cut into pieces and smashed with liquid nitrogen. Total RNAs from leaflets were prepared using Trizol reagent (Thermo Fisher, Waltham, MA, USA). The supernatants containing total RNAs were then purified using Direct-zol RNA MicroPrep Kit (Zymo Research). The concentration of total RNAs was measured using NanoDrop (Thermo Fisher, Waltham, MA, USA) and 0.1–1 μg of total RNAs were reverse transcribed with cDNA Synthesis Kit (Thermo Fisher, Waltham, MA, USA) in accordance with the manufacturers' instruction. The cDNAs were further sixfold diluted

and subjected into RT-qPCR experiments using QuantStudio 6 Real-Time PCR Detection System (Thermo Fisher; Waltham, MA, USA) and KAPA SYBR FAST qPCR Master Mix (2X) Kit (Kapa Biosystems, Wilmington, DE, USA). Primers for RT-qPCR were designed using Primer 5.0 software and synthesized by Invitrogen. Expression was normalized to GAPDH. The data were analyzed using the 2-$\Delta\Delta$CT method.

## Statistical analysis

All results were analyzed by SPSS 22.0 software (SPSS lnc, Chicago, IL, USA). The Chi-square test was used to compare the count data between two groups. One-way ANOVA was used for comparation between multiple groups. Tukey test for multiple comparisons was used to generate the $P$ value between two groups in one-way ANOVA. Data were shown as mean ± SD and displayed using GraphPad Prism version 8 (GraphPad Software Inc., San Diego, CA, USA). The $P$ value < 0.05 was considered statistically significant.

## RESULTS

### The construction of FEM of aortic valve and ascending aorta

The image data of four participants, including two BAV patients and two healthy volunteers were used to generate 3D geometric model of aortic valve. The shape of aortic valve and the cardiac function of four participants were evaluated using cardiac ultrasonic test (Fig. 1A). These two patients were diagnosed with left–right (RL) leaflet fusion BAV and the function of aortic valve was normal. The data of ascending aorta CTA examination were used to construct 3D geometric surface model (Fig. 1B).

The 3D aorta geometric surface model was generated based on a series of CT angiography images, and the final geometry included the intact aortic valve as well as ascending aortic arch. The structure of heart valve is similar to the thin-walled leaflet, and bending stress is the main cause of valve damage during cardiac cycle (*Weinberg et al., 2010*). Shell element was used in our study to construct the FEM of aortic valve since bending stress was taken into consideration in it (*Kim et al., 2007*), and linear hexahedral element was used to generate the FEM of ascending aorta (Fig. 2A). Further, based on the tissue density of heart valve, the 3D geometric model of BAV (Fig. 2B) and TAV (Fig. 2C) were constructed and used for equivalent stress analysis.

### Dynamic simulation of the aortic valve

For BAV patients, the abnormal hemodynamics at the aortic valve can cause the valvular calcification. More evidence indicated that mechanical stimulation is the main cause of cardiac valve calcification (*Bogdanova et al., 2018*). Based on the FEM, we investigated the equivalent stress on different part of heart valves during one heart cycle. Dynamic FE analyses of the aortic valve model during one heart cycle under different pressure were performed.

At the across valve pressure difference of 25mmHg, von Mises stress distribution on the valve leaflets during the opening or closing phase were shown in Fig. 3A. At the fully open position, the equivalent stress at the belly of the BAV large leaflet was higher than that of TAV. At the closing phase, there was a local high stress zone formed at the belly of
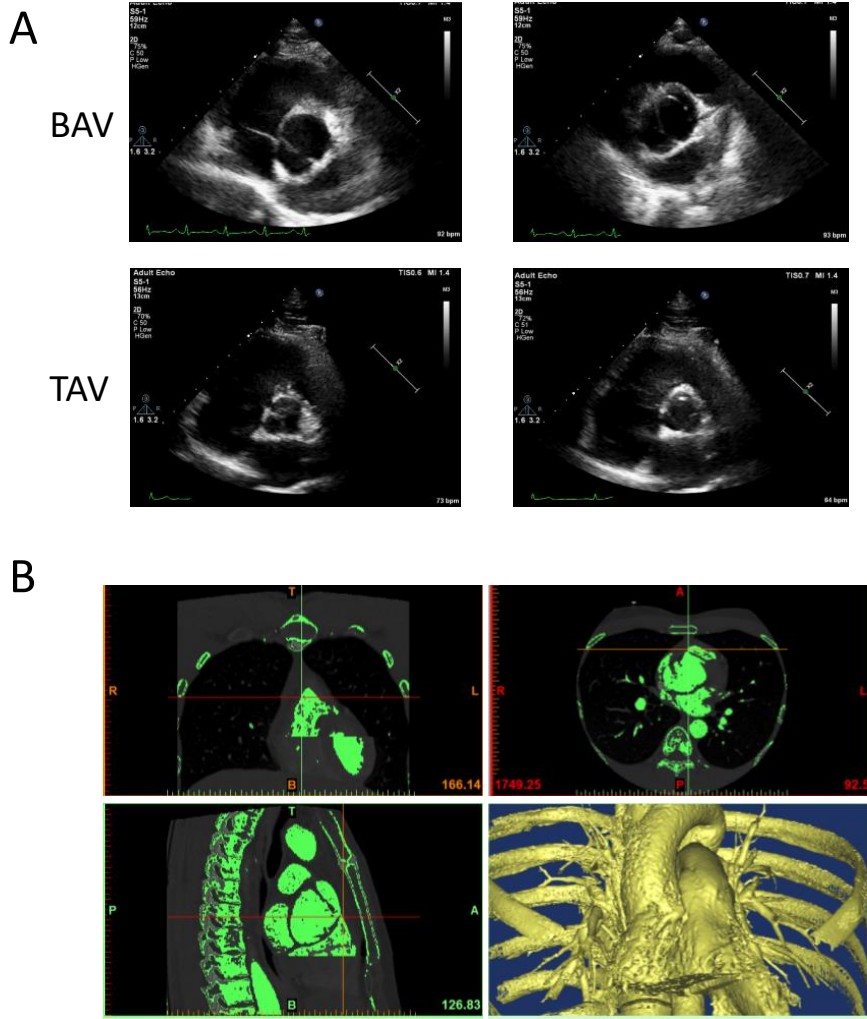

**Figure 1  The construction of 3D geometric surface model of aortic valve and ascending aorta.** (A) The cardiac ultrasonic image of congenital bicuspid aortic valve patients (top) and tricuspid aortic valve volunteers (bottom). (B) The 3D model of aortic valve based on the ascending aorta CTA image.

BAV large leaflet. In order to investigate the mechanical stimulation at different parts of leaflet, we draw the time-dependent aortic pressure curve during one cardiac cycle (Fig. 3B). There was a stress peak at the systolic phase and the equivalent stress of different parts of the heart valve had no significant difference. At diastole phase, the peak values of BAV large leaflet belly, BAV large leaflet free edge, BAV small leaflet belly, BAV small leaflet free edge, TAV leaflet belly and TAV leaflet free edge were 102 KPa, 54 KPa, 52 KPa, 50 KPa, 45 KPa and 60 KPa respectively.

A similar phenomenon was observed at the across valve pressure of 50mmHg. Overall, the stress on aortic leaflet increased with the elevation of across valve pressure. At the diastole phase, the BAV large leaflet belly suffered higher stress compared with other parts of BAV and TAV (Fig. 4A). The time-dependent stress curve showed that the peak values

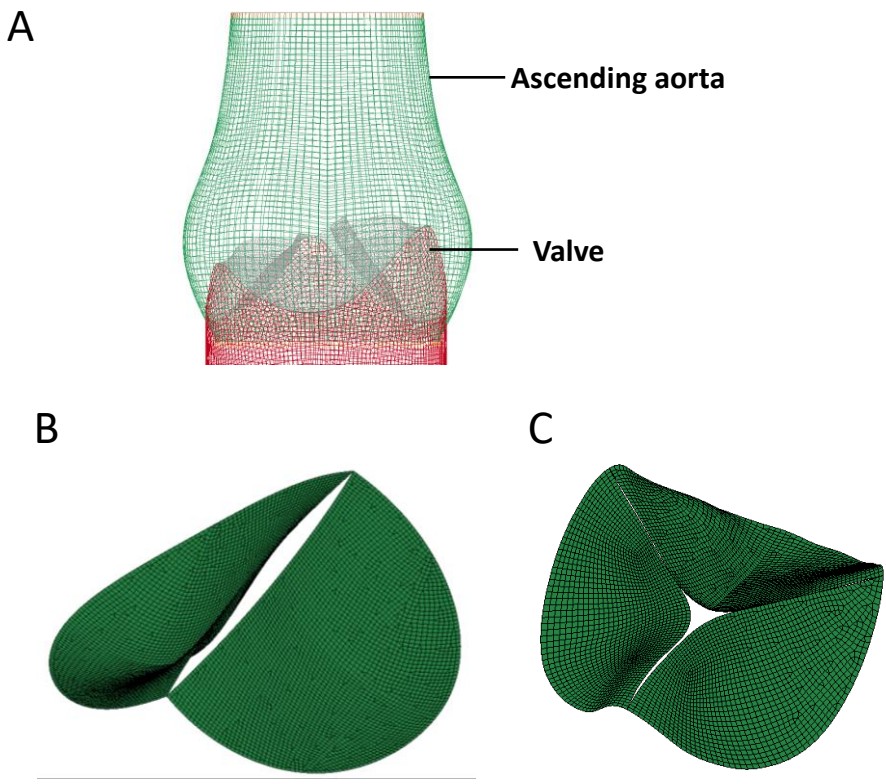

**Figure 2** **The construction of FEM model of aortic valve.** (A) Geometric model of the aortic valve and ascending aortic arch. (B) Boundary setting for the bicuspid aortic valve. (C) Boundary setting for the tricuspid aortic valve.

of BAV large leaflet belly, BAV large leaflet free edge, BAV small leaflet belly, BAV small leaflet free edge, TAV leaflet belly and TAV leaflet free edge were 195 KPa, 100 KPa, 73 KPa, 98 KPa, 105 KPa, and 115 KPa, respectively, at the diastole phase (Fig. 4B).

In general, with the same valve pressure, the difference between BAV and TAV leaflets was not significant at the systolic phase. The equivalent stress on BAV was higher on TAV leaflet at the diastole phase, especially on BAV large valve belly.

## Notch1/NICD/Runx2 signaling pathway participate in the calcification of BAV

At diastole phase, the oscillatory shear stress on fibrosa layer of leaflet can promote the valvular calcification (*Back et al., 2013*). The Notch1 signaling pathway plays an important role in the pathological process of CAVD. Based on the findings of FEA, the higher stress on BAV large valve belly might cause disturbance of Notch1 pathway, leading to the calcification of BAV. We collected the aortic valves from BAV and DCM patients with TAV and their clinical characteristics were shown in Table 1. There was no significant difference of age, gender and CAVD risk factors between these two groups. ECG parameters differed between BAV patients who had valvular calcification and DCM patients with heart failure.

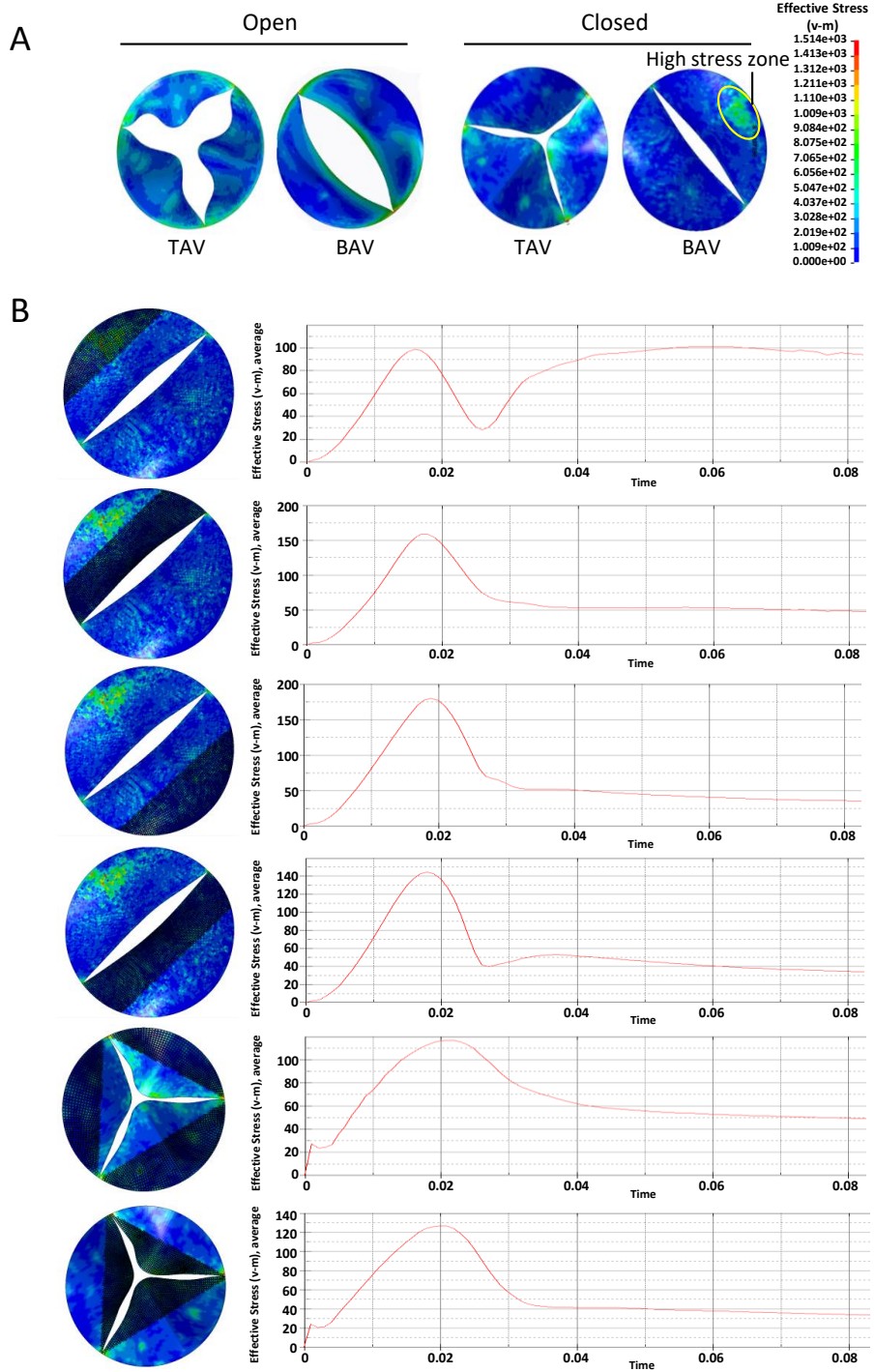

**Figure 3  Analysis of the equivalent stress on heart valves during the cardiac cycle at the pressure difference of 25 mmHg.** (A) Equivalent stress distribution on the valve leaflets during the opening or closing phase. (B) The time-dependent aortic pressure curve at different part of BAV and TAV during one cardiac cycle.

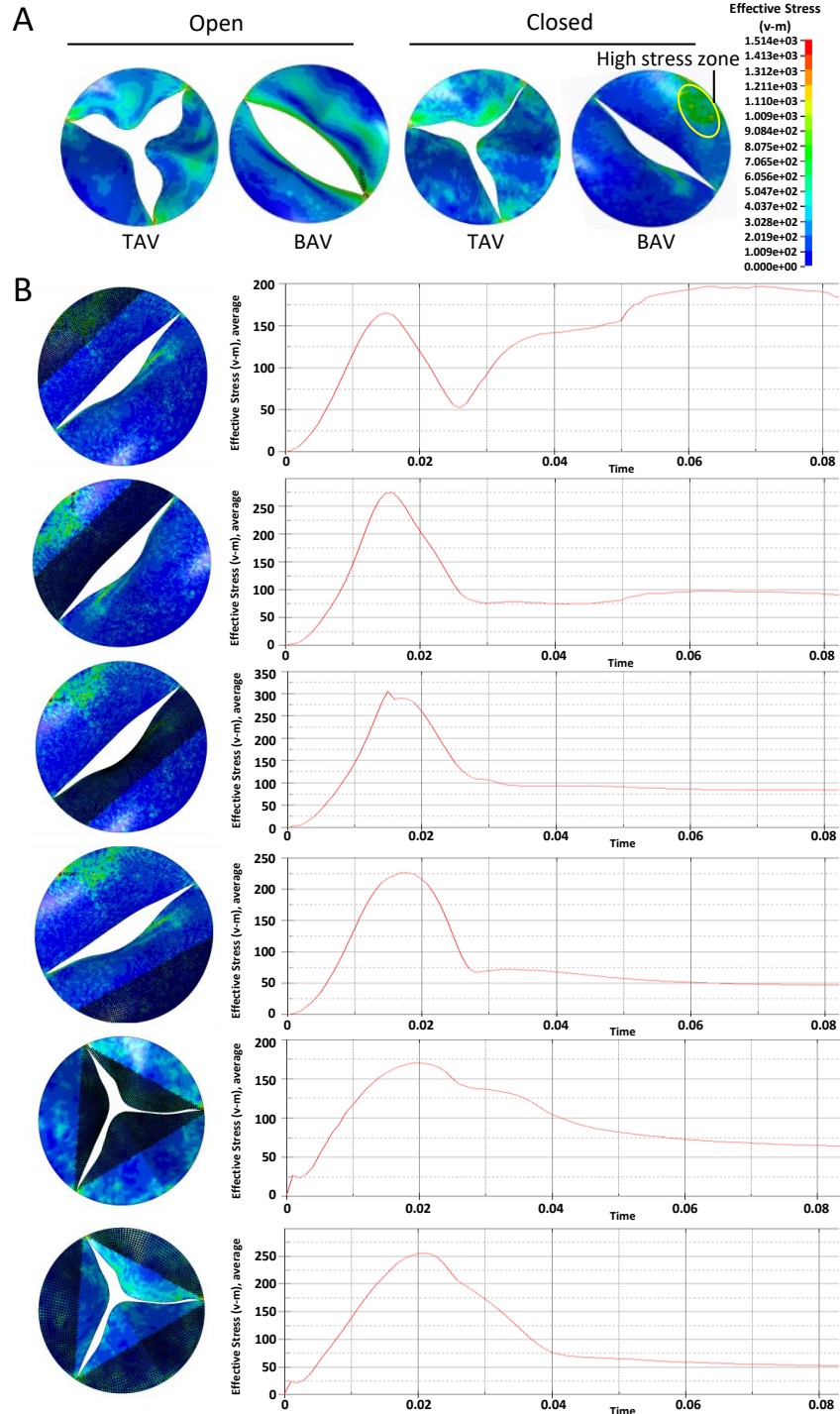

**Figure 4 Analysis of the equivalent stress on heart valves during the cardiac cycle at the pressure difference of 50 mmHg.** (A) Equivalent stress distribution on the valve leaflets during the opening or closing phase. (B) The time-dependent aortic pressure curve at different part of BAV and TAV during one cardiac cycle.

We further explored the expression level of molecules in the Notch1 pathway at different part of BAV and TAV leaflets (Fig. 5A). Alizarin red staining showed that large BAV had significantly more calcification nodules compared with small BAV and TAV (Fig. 5B). The expression of Notch1 on large BAV leaflet belly was significantly lower than that on small BAV leaflet and TAV, which is negatively associate with equivalent stress on leaflet. The activation of Notch1 mediates the release of NICD, and the activation of NICD can inhibit the expression of Runx2, which mediates the calcification of aortic valve (*Toshima et al., 2020*). The expression pattern of NICD on different parts of aortic valve corresponded with Notch1, and the level of Runx2 on large BAV leaflet was significantly elevated as insufficient of NICD (Figs. 5C–5D). We also extracted the mRNA from different parts of aortic valve, and we found the mRNA of Notch1 was significantly lower at large BAV leaflet belly, on the contrary, the level of pro-calcification gene Runx2 was significantly higher at the root of large BAV (Fig. 5E).

## DISCUSSION

The aortic valve is in a dynamic cycle of opening and closing, and its mechanical stimulation also changes consistently. Studies indicate that FEA is an effective method to simulate the complex hemodynamic environment of aortic valve and analyze the mechanical stress on leaflet (*Sun, Chandra & Sucosky, 2012*; *Schipke, To & Warnock, 2011*; *Wang et al., 2021a*). The construction of 3D models to display accurate morphological and mechanical properties of aortic valves is extremely important in FEA. Several studies imported MRI data to construct 3D model (*Jernigan et al., 2007*). We could also obtain multilevel scanning data from MRI test, but we found that the resolution of soft tissues was very high in MRI scanning. The MRI images of aortic valve were mixed with myocardial tissue and soft tissue artifacts, which made it difficult to construct the solid structure of aortic valve. Cardiac color doppler ultrasound is widely used in the diagnosis of BAV, but it is hard to control the scanning layer thickness. In our study, we constructed FEM of aortic valve using CTA images. This method can scan aortic valves in multiple levels at different cardiac cycles. And the injection of contrast agent can enhance the ascending aorta imaging, distinguish leaflet from myocardial tissue and soft tissue. In general, the CTA images were more appropriate in the construction of our 3D model.

Based on previous studies, blood flow at BAV showed asymmetrical distribution. On the contrary, blood flow at TAV was symmetrical in each aortic segment (*Viscardi et al., 2010*). Thus, different parts of BAV might suffer from various mechanical stress due to the asymmetrical distribution of flow. In our study, we compared the morphological and mechanical characteristics between BAV and TAV leaflets. The special shape of BAV belly caused the abnormal movement of valve during cardiac cycle. The leaflets of BAV could not open completely, creating an eccentric oval opening and resulting in aortic valve stenosis. The large BAV covered more valve opening area during diastole phase, lead to the budge of BAV body, which means that BAV would suffer more bending stress than TAV. We investigated the equivalent stress on valves during one cardiac cycle, and the result showed that the large BAV belly bears higher equivalent pressure, which might promote the

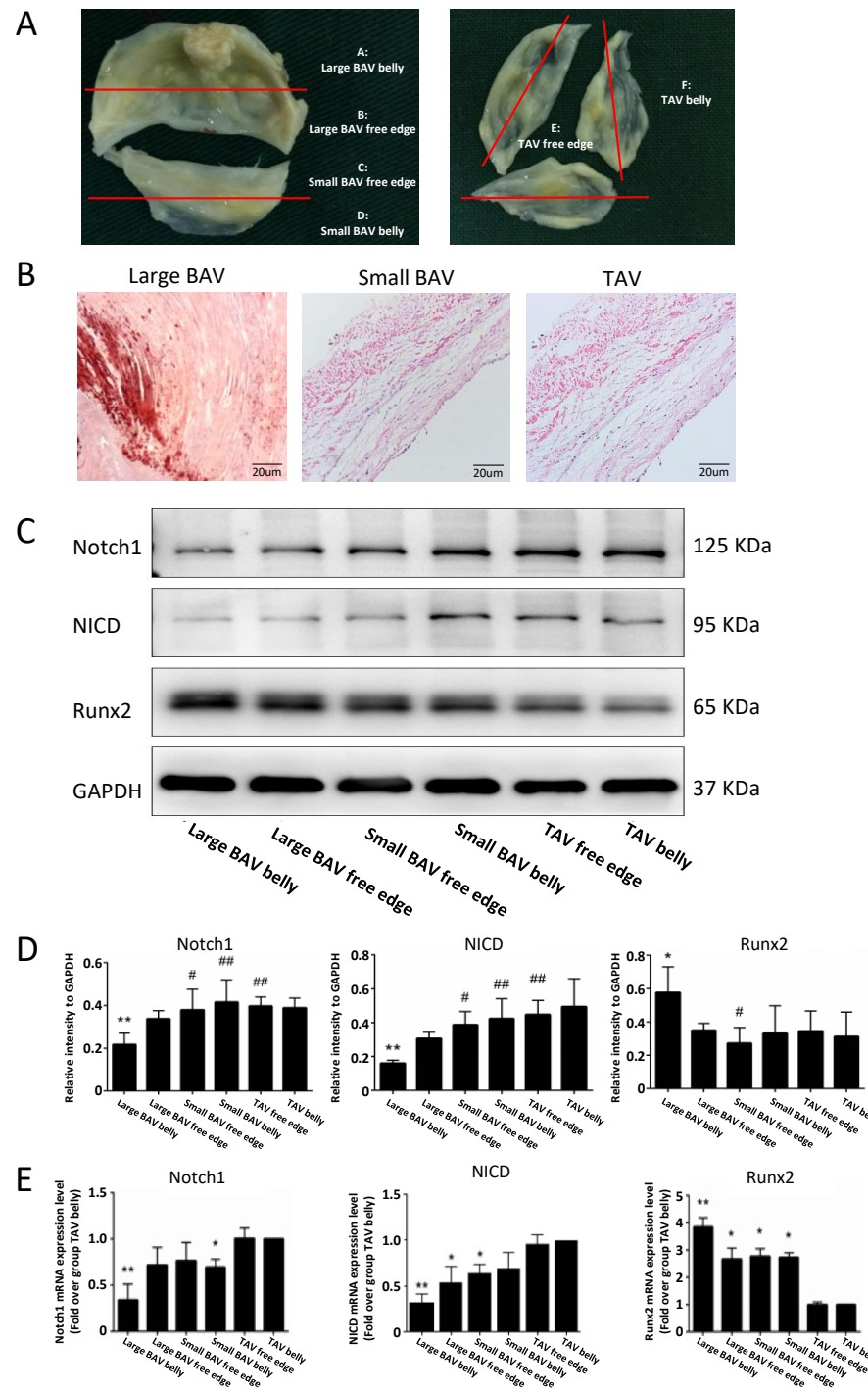

**Figure 5** **Notch1/NICD/Runx2 signaling pathway participant in the valvular calcification.** (A) Samples collected from different parts of BAV and TAV were distributed into group A–F. (B) Calcification nodules stained by alizarin red on large BAV, small BAV and TAV. (C) The protein levels of Notch1, NICD and Runx2 at different parts of aortic valve in immunoblot assay. (D) The bar plots displayed the protein (continued on next page…)

**Figure 5 (...continued)**
levels of Notch1, NICD and Runx2 at different parts of the aortic valve. Data are mean ± SD ($N = 3$) from three independent experiments. *$p < 0.05$ compared with group TAV belly; **$p < 0.01$ compared with group TAV belly; #$p < 0.05$ compared with group large BAV belly; ##$p < 0.01$ compared with group large BAV belly. (E) The bar plots showed the mRNA level of Notch1, NICD and Runx2 at different parts of the aortic valve. Data are mean ± SEM ($N = 10$) from three independent experiments. *$p < 0.05$ compared with group TAV belly; **$p < 0.01$ compared with group TAV belly.

pathogenesis of secondary complications. Furthermore, we found that large BAV had more calcification nodules, indicating that the calcification was correlated with the disturbance of mechanical stress.

Multiple studies indicated that BAV-associated hemodynamic alterations were likely to contribute to the calcification of aortic valves. *Qin et al. (2020)* found a strong correlation between regions with a high risk of calcification and regions with elevated mechanical stress. However, the changes of intracellular signaling cascades caused by abnormal mechanical stress still needed more investigation (*Kazik et al., 2021*).

Notch signaling was one of the most important pathways to maintain tissue morphogenesis and homeostasis. Notch1 mutations are the first identified human genetic variants that cause BAV (*Garg et al., 2005*). Several studies showed that Notch1 signaling pathway participate in the pathogenetic process of CAVD (*Gao et al., 2022*). The activation of Notch1 could release NICD to cell nucleus (*Zhou et al., 2019*). Runx2 is the target gene of NICD and the activation of NICD can inhibit the expression of Runx2. Runx2 is a marker gene of osteoblast like-cell, which mediate the calcification of aortic valve (*Dharmarajan et al., 2021*). In our study, we investigated the expression of Notch1, NICD and Runx2 at different parts of BAV and TAV. We found the level of Notch1 and NICD were significantly lower at the root of large BAV, where had higher equivalent pressure. On the contrary, the expression of Runx2 at large BAV belly was significantly higher. Therefore, we concluded that high mechanical pressure at large BAV belly might disturb Notch1/NICD/Runx2 signal pathway.

However, our study has some limitations. Multiple molecules could participate in the progression of CAVD and Notch1 signal might be one of them. Comparing the transcriptome and proteome among different parts of aortic valve can provide more information about the signaling pathway involved in BAV-associated calcification. In addition, to dynamically investigate the progression of calcification, we need to evaluate the changes of aortic valves at different stages of CAVD.

In our study, we preliminarily verified the correlation between the Notch1 signaling pathway and mechanical stress on aortic valve. The morphological deformity of BAV caused the disturbance of the hemodynamic environment, and the abnormal mechanical pressure on BAV could interfere with the Notch1/NICD/Runx2 signal pathway. There is still much work to do to investigate the mechanism during valvular calcification and figure out the strategies to prevent the progression of CAVD.

### Funding

This work was supported by the National Natural Science Foundation of China (No. 31500794). The funders had no role in study design, data collection and analysis, decision to publish, or preparation of the manuscript.

### Grant Disclosures

The following grant information was disclosed by the authors:
National Natural Science Foundation of China: 31500794.

### Competing Interests

The authors declare there are no competing interests.

### Author Contributions

- Guangzhou Li conceived and designed the experiments, performed the experiments, analyzed the data, prepared figures and/or tables, and approved the final draft.
- Na Shen conceived and designed the experiments, performed the experiments, analyzed the data, authored or reviewed drafts of the article, and approved the final draft.
- Huifang Deng conceived and designed the experiments, prepared figures and/or tables, and approved the final draft.
- Yixuan Wang conceived and designed the experiments, performed the experiments, analyzed the data, authored or reviewed drafts of the article, and approved the final draft.
- Gangcheng Kong analyzed the data, authored or reviewed drafts of the article, and approved the final draft.
- Jiawei Shi analyzed the data, prepared figures and/or tables, and approved the final draft.
- Nianguo Dong analyzed the data, authored or reviewed drafts of the article, and approved the final draft.
- Cheng Deng analyzed the data, authored or reviewed drafts of the article, and approved the final draft.

### Human Ethics

The following information was supplied relating to ethical approvals (*i.e.*, approving body and any reference numbers):

This research was approved by the Union Hospital and Tongji Medical College (Ethical Application Ref: IORG0003571).

### Data Availability

The raw data are available in the Supplementary Files.

### Supplemental Information

Supplemental information for this article can be found online at http://dx.doi.org/10.7717/peerj.14950#supplemental-information.

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
