# Peer review of "Abnormal mechanical stress on bicuspid aortic valve induces valvular calcification and inhibits Notch1/NICD/Runx2 signal"

_PeerJ, doi:10.7717/peerj.14950_

## Round 0.1 · original submission · Major Revisions

Your paper has been assessed by three independent reviewers and I strongly suggest addressing the concerns raised by all three reviewers before your paper can be considered for publication.

1.Introduction has to be reframed with enough background information, the rationale and novelty of the current study.

2. In the methodology section, authors are recommended to provide more information on the sources of products used for the study.

3. The authors are recommended to elaborate on the implications of the current study in the discussion instead of repeating the results.

4. The amount of molecular profiling and signaling pathway studied in the current study does not justify the given title. The authors are advised to change the title.

Reviewer 1 ·

Basic reporting

The authors demonstrate the effect of hemodynamic/mechanical stress in promoting calcification at the bicuspid valve by dysregulated Notch1 signalling between regions of the leaflets. They have given detailed reasoning for the experimental methods on proceeding with FEA from CTA rather than other methods like MRI and showed data that validate their method.

Experimental design

No comment

Validity of the findings

Looking at the image of the BAV and TAV in Fig.5A and the subsequent expression profile of the Notch1 pathway proteins, a histological assessment of the belly of BAV 1) showing presence of calcification and 2) colocalised with expression difference in Notch1/RUNX2 between the belly and other regions would strength their conclusion.

Additional comments

Minor comments:
1. Please remove the "kind of" in the first line of abstract.
2. Line 41-43 needs to be reframed for better understanding.

Reviewer 2 ·

Basic reporting

The over all language of the manuscript is easy to understand and professional. Though at few places the sentence formations are a bit long. Long sentences are difficult to follow and most certainly dilute the message the authors want to convey.
Eg: Lines 41-43 of "Introduction" and Lines 247-250 of "Discussion"

The literature reference in the "Introduction" section is as it should be. In "Discussion" section, the review should be more comprehensive. The results' discussion is not connected well to the available data/literature.

The raw data for figure 1 and 2 are missing. For figure 5, the authors have provided the cropped images of the western blot. It is recommended to provide the images of full blots with marker lane. Same goes for the excel sheet of Fig. 5. The sheets are unlabeled (with only values for specified groups written down) and therefore hard to follow what is being shown there.

Experimental design

1) The experimental design though not novel is well defined and relevant. The work tries to identify the gap in the role of Notch1/NICD/Runx2 signaling pathway on BAV calcification. But as per my understanding it should be more detailed.

2) In "material and method" section under sub-section "Antibodies and primers" the primers for NICD are not provided. It is advised to check the mRNA of NICD as well, given the scope of manuscript.

Validity of the findings

1) The raw data provided is not robust with data for certain figures altogether missing.

2) Fig 5 also needs mRNA analysis of NICD since the authors are focusing on the role of Notch1/NICD/Runx2 pathway

3) One major aspect missing from the manuscript is the effect (if any) of the different parameters shown in table 1. For example how could, the risk factors or the medications of the patients recruited in the study, change the outcome? The authors are recommended to incorporate it in the manuscript.

Additional comments

The manuscript title focuses on the Notch1/NICD/Runx2 signaling pathway but the experiments are less focussed on it. Hence it is somewhat mis-leading. The authors are advised to work on that. Because when talking about signaling pathways a more detailed molecular profiling is needed.

Reviewer 3 ·

Basic reporting

None

Experimental design

In the manuscript, Li et. al. used the finite element analysis to investigate the role of mechanical stress on BAV and aortic calcification and the role of Notch1 signaling in the process. Though the results are interesting, I have a limitation for the results:
Major:
• It was difficult for me to understand the significance of the work. FEA has been previously used for BAV, which has predicted similar outcomes, and the involvement of the Notch1 process in the same is also known. Evidence for the presence of aneurysms and abnormal hemodynamics at the aortic leaflets is also there in the literature. The definite impact of the paper is not coming out.
• The introduction of the manuscript is jumbled up. The authors have failed to explain a linear rationale for their study. The introduction needs to be reframed.
• The discussion does not appropriately signify the study. The authors have just elaborated on their results, without highlighting the impact.
Minor:
• Correct the grammar in the introduction line 41-43.
• The authors should consider including the reference for FEA, maybe in the context of other heart-related diseases in the introduction section.
• What is the role of Notch1 in the normal structure and functioning of the aortic valve? The authors should include this in the introduction/discussion sections.
• In line 79-81, “the clinical characteristics…” the references for the specific table or the figure is lacking.
• Is there a reference for the aortic valve geometric model? If yes, please include it in the methods section.
• The text on all the geometric models and the graphs are either is too small or blurry. This could be improved.
• What is the N for the western blots and the rt-PCR? Kindly mention it in the legends. The western blots should be accompanied by a quantification. The A, B, C…. F pattern of labeling for the blots is difficult to read. The authors should consider labeling it beside the blot for better understanding. The same is also true for the rt-PCR data.

Validity of the findings

None

Additional comments

None

---

## Round 0.2 · Minor Revisions

The authors have addressed the comments from the reviewers. But still, the complete Western blots are unavailable. The authors are required to provide full Western blot images before being considered for publication.

Reviewer 1 ·

Basic reporting

The authors have greatly improved the manuscript and the methods and results support the conclusions of the study. It can be accepted as is for publication

Experimental design

No comments

Validity of the findings

No comments

Reviewer 2 ·

Basic reporting

Authors have addressed majority of issues raised in prior review.
But a major concern that still remains is the unavailability of complete western blots in raw data. For results to be convincing and reliable enough the complete blots are a must.

Experimental design

NA

Validity of the findings

NA

Reviewer 3 ·

Basic reporting

None

Experimental design

None

Validity of the findings

None

Additional comments

None

---

## Round 0.3 · Minor Revisions

The authors have provided the Western blot images but the molecular weight of NCID protein in the manuscript and the raw data file seem to be inconsistent. Also, there is a spelling mistake in the "NOTCH 1" protein in the raw data file. The authors are asked to fix these errors and resubmit.

---

## Round 0.4 · accepted · Accept

The authors have addressed the comments and the manuscript is ready for publication.